# Isolation and Cultivation of Human Gut Microorganisms: A Review

**DOI:** 10.3390/microorganisms11041080

**Published:** 2023-04-20

**Authors:** Xuchun Wan, Qianqian Yang, Xiangfeng Wang, Yun Bai, Zhi Liu

**Affiliations:** Department of Biotechnology, College of Life Science and Technology, Huazhong University of Science and Technology, Wuhan 430074, China; m202272387@hust.edu.cn (X.W.); d202280859@hust.edu.cn (Q.Y.); d202080696@hust.edu.cn (X.W.)

**Keywords:** gut microbes, human, culturomics, targeted sorting, enrichment, uncultured microbes

## Abstract

Microbial resources from the human gut may find use in various applications, such as empirical research on the microbiome, the development of probiotic products, and bacteriotherapy. Due to the development of “culturomics”, the number of pure bacterial cultures obtained from the human gut has significantly increased since 2012. However, there is still a considerable number of human gut microbes to be isolated and cultured. Thus, to improve the efficiency of obtaining microbial resources from the human gut, some constraints of the current methods, such as labor burden, culture condition, and microbial targetability, still need to be optimized. Here, we overview the general knowledge and recent development of culturomics for human gut microorganisms. Furthermore, we discuss the optimization of several parts of culturomics including sample collection, sample processing, isolation, and cultivation, which may improve the current strategies.

## 1. Introduction

Numerous studies have shown that gut microbes are important for host health [1,2,3,4,5]. For example, *Clostridium difficile* colonization in the intestine tends to cause diarrhea [6], while intestinal commensal bacteria can strengthen the intestinal barrier and help to resist its colonization [7]. Culture-dependent and culture-independent approaches provide multiple perspectives on the understanding of gut microbes and their relationship with the host, which allow for the continuous improvement of the related information base [8,9,10,11]. At the same time, a library of human intestinal strains for subsequent empirical studies, product development, and bioinformatics annotation has also been gradually established [12,13]. After culturomics was proposed, this pool of strains was significantly expanded, and many microorganisms previously overlooked or considered unculturable were brought into cultivation [14]. Many researchers have realized that it is possible to culture human gut microbes at a higher level [15]. More importantly, with the elucidation of some novel strains’ beneficial mechanisms, mining the next generation of probiotic candidates from the human gut has become a promising research hotspot [16,17,18,19]. Thus, culturomics is gaining more and more attention [20]. 

To improve the efficiency of obtaining human gut microorganisms, researchers have attempted to optimize some parts of culturomics [21,22,23]. Advanced molecular tools and single-cell techniques have also been attempted to combine traditional isolation with culture methods [24,25]. As strategies to obtain microbial strain resources do not apply exclusively to human gut microbes [26,27,28], the strategies used in non-intestinal habitats may need to be tailored to the characteristics of human intestinal microorganisms. Furthermore, developing new approaches by coupling multiple techniques is worth trying [23]. In this paper, we analyze the strategies used to obtain human gut microbial resources. We also discuss the optimization of parts of culturomics including sample collection, sample processing, microbial isolation, and cultivation, aiming to provide a possible reference for developing better approaches to obtaining microbial resources in the human gut.

## 2. Culturomics: Nontargeted and Targeted Strategies

Culturomics is a strategy for microbial isolation and cultivation, which was introduced by the group of Didier Raoult and Jean-Christophe Lagier in 2012 [29]. This strategy has enriched the information base of microbes from humans, such as the matrix-assisted laser desorption ionization time-of-flight mass spectrometry (MALDI-TOF) database and a microbial gene bank. More importantly, it has provided microbial entities for subsequent studies at a biochemical level, as well as cultured several microbes that were neglected due to low abundance or biased culture-independent analysis [30]. As of 2021, the species of bacteria obtained from humans reached 3253 with an addition of 477 species compared to the 2018 statistics. Of this increment, 63% were contributed by culturomics [14]. Similarly, 66.2% of the new species added in 2018 relative to 2015 were obtained through culturomics [31]. 

In general, the procedure of culturomics contains several steps, including sample collection, sample processing, microbial isolation, cultivation, identification, and preservation. Typically, classic culturomics is a nontargeted strategy. Processed samples are spread on several Petri dishes with different media. After culturing for several days, the microbial colonies of interest are picked and then identified using MALDI-TOF. If a reference spectrum is lacking, 16S rRNA sequencing needs to be performed for further identification. In detail, sequencing results are compared with the nearest evolutionary-related species in the database, and, if the similarity is <98.5%, the species can be considered new. Next, these new species are characterized, classified, and applied (Figure 1) [30]. The nontargeted strategies are more likely to isolate and culture microorganisms as much as possible.

Additionally, targeted strategies can also enhance access to gut microbial resources [32]. The development of molecular biology, bioinformatics, flow cytometry, and microfluidics techniques have provided the technical basis for the targeted sorting of gut microbes [24,33,34,35,36,37,38]. Except for the same procedures with the nontargeted approach, researchers need specific information on microorganisms for targeting, which is usually obtained through literature research, metagenomics data, etc. In combination with high-throughput cell sorters, targeted strategies may be more conducive to obtaining microorganisms of interest [39]. Effective labeling techniques and advanced screening devices are urgently required for targeted approaches, which may challenge low-resource labs. Therefore, researchers should choose appropriate strategies on the basis of their experimental resources and goals, regardless of a nontargeted or targeted approach. 

Despite the significant contribution of culturomics in human gut microbial isolation and cultivation, there are still some constraints. For instance, the nontargeted strategies are labor-intensive and may not necessarily result in the capture of the wanted taxa and cause a large waste of manpower and resources [40]. For targeted strategies, some huge sorters may be hard to set up in an anaerobic workstation, which may hinder the isolation and cultivation of certain microorganisms. Some microorganisms may have died before entering the culturomics procedure [33]. Some symbiotic relationships among microorganisms have not been clarified [41]. Accordingly, culturomics has been continually optimized in different parts of its procedure. 

## 3. Optimizations of Culturomics

### 3.1. Sample Collection (Figure 2)

Collecting feces with tubes in vitro has been the most commonly used sampling method due to its convenience, economic benefits, and reproducibility [42]. If samples are sent for sequencing, FTA cards and other sampling kits are available on the market, which can adequately maintain DNA [43,44]. Despite the advantages of the ‘fecal way’, the fecal microbiota is not equally distributed within feces and has its biostructure, which suggests that the microbial composition of two samples collected from the same stool may differ significantly [45]. There may be significant differences in microbial composition between the intestinal mucosa and feces [46,47], which may allow for distinction in the microbial composition among fecal, fluid, and biopsy samples [46]. Microorganisms associated with the mucosa may be lost through the ‘fecal way’. 

In addition to the ‘fecal way’, endoscopic approaches may be a good alternative to access more microbial resources, which provide a relatively direct route for sampling gut microbes in vivo [48]. With an endoscope, researchers can use tools such as forceps, aspirators, and brushes to accurately collect the contents of specific areas of the intestine, including fluid and biopsy [49,50,51]. Despite the high accuracy of this method, it has insurmountable disadvantages, such as high cost, cumbersome processes, and inevitable contamination. Bowel preparation, a necessary procedure for endoscopic operation, may disrupt the original composition of gut microbes. Although the endoscopy-assisted approach helps to tap into microorganisms associated with the intestinal mucosa, it is still far from realistic to promote among healthy people.

In recent years, a promising collection tool has been intelligent capsules. For example, Rezaei et al. [52] developed an ingestible, biocompatible, battery-less, 3D-printed micro-engineered pill for in vivo sampling in the small intestine. The system contains an integrated osmotic sampler and microfluidic channel, which can aspirate intestinal fluid. Another swallowable device for sampling consists of a capsule system and a quencher. The capsule system containing sensors and a power unit can be initiated by specific pH to aspirate intestine fluid into the quencher [53]. In addition, Zhen et al. [54] designed a convenient, noninvasive, and accurate sampling capsule called a magnetically controlled sampling capsule endoscope (MSCE). This capsule can noninvasively and accurately acquire both jejunal and ileal gastrointestinal (GI) content via direct visualization under magnetic control. Compared to the fecal and endoscopy-assisted approach, the intelligent capsule can precisely collect fluid in the different intestine sites while avoiding the negative effects of the invasion. However, this method is not widely used because of its high technical barriers and costs. 

**Figure 2 microorganisms-11-01080-f002:**
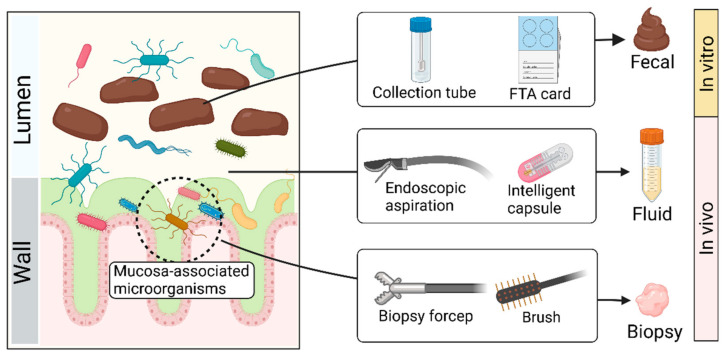
Schematic diagram of sample collection. Created with BioRender.com.

### 3.2. Sample Processing (Figure 3)

#### 3.2.1. Microbial Enrichment

Raw samples from humans often require appropriate processing to facilitate both culture-independent and culture-dependent studies of gut microbes. An important procedure is microbial enrichment. Several studies have demonstrated the beneficial effects of microbial enrichment on access to microbial resources [55,56]. The basic principle of enrichment culture is to set one or more restrictive conditions on a mixed microbial sample according to the characteristics of the microorganisms so that the relative or absolute abundance of the eligible microorganisms is enhanced. Differences in the physical characteristics of microorganisms, including buoyancy density, sedimentation coefficient, size, morphology, and refractive properties, have all been successfully applied to enrich certain microorganisms. In addition, selective media are also used for microbial enrichment. In detail, physical conditions, nutrients, antibiotics, and phages can be adjusted to the design of selective mediums [57,58,59,60,61]. Some slow-growing bacteria can be eliminated by microbes with growing advantages, due to competitive factors, which may result in the relative or absolute abundance of some intestinal microorganisms being too low to be eventually isolated and cultured in vitro [57]. In this case, one of the strategies to assist in isolating such slow-growing bacteria is to enrich them with nutrient-poor media [62]. Another strategy to protect bacteria from other microorganisms or environmental pressure is droplet-based approaches. For example, Yin et al. [63] prepared single-bacterium droplets with a high-throughput microfluidic device. Next, the droplets were cultured on agar plates to form discrete single-cell colonies. Then, using the conditioned plates containing metabolites from the engineered butyrate-producing bacteria (EBPB) supernatants, they obtained gut bacteria closely associated with or interacting with the EBPB. This droplet-based method ensured good microbial diversity and abundance. Notably, it could be operated in an anaerobic chamber, showing great potential for capturing anaerobic gut microbes. Ethanol was also used for sample processing to suppress the growth of dominant species so that slow-growing bacteria could survive. Furthermore, a study by Chang et al. suggested that adding fresh medium and extending the enrichment time are beneficial to isolate more bacterial species with less work. Choosing optimized timepoints for inoculation can also significantly increase the number of isolated strains [22]. In conclusion, enriching certain microorganisms according to demand can improve the efficiency of isolating them.

#### 3.2.2. Anaerobic Protection

Since the gut is an anaerobic environment, the impact of oxygen on microbial diversity needs to be considered during processing [64]. In general, deaerators, anaerobic chambers, and anaerobic jars are used to protect oxygen-sensitive intestinal microorganisms [64,65,66]. However, in some cases, it is hard to set up sorters or other devices under anaerobic chambers. Interestingly, Jeevarathinam et al. [67] constructed a simple, cost-effective, and scalable microparticle system containing glucose oxidase and catalase, which could precisely regulate dissolved oxygen concentration via glucose oxidase-mediated consumption of oxygen. The system can be used as a stimulated gut in vitro, which helps to culture gut microorganisms. They tested the effectiveness of this system with obligate anaerobe *Bacteroides thetaiotaomicron* and found growth rates comparable to an anaerobic chamber, providing an alternative for low-resource laboratories.

#### 3.2.3. Sample Preservation

Appropriate preservation can maintain the originality of samples, regardless of sequencing or cultivation. Normally, samples for cultivation should be immediately used for microbial isolation or be refrigerated at 4 °C for a short time. However, for longer storage, they should be stored in a −80 °C refrigerator [68,69]. Samples for sequencing can also be stored in a −80 °C refrigerator. However, in some conditions where quick-freezing is not convenient, samples for sequencing can also be stored by other options such as using 95% ethanol and RNAlater, fecal occult blood test cards, FTA cards, and OMNIgene Gut to preserve feces at room temperature for several days [40,44,70,71,72]. Although sequencing is just an indirect factor to guide microbial isolation and cultivation, it is necessary to minimize the impact of sample preservation on the accuracy of sequencing, which may benefit culturomics.

**Figure 3 microorganisms-11-01080-f003:**
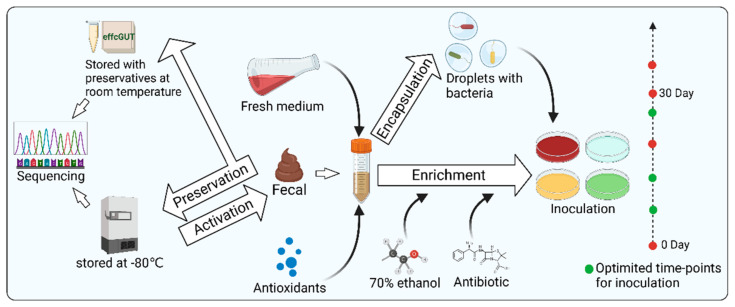
Schematic diagram of sample processing including microbial enrichment, anaerobic protection, and sample preservation. Created with BioRender.com.

### 3.3. Microbial Isolation

To isolate pure culture, processed samples are usually spread on Petri dishes containing different media to form single colonies. Then, those colonies are to be further purified and identified. Recently, Yadav et al. [41] reported a novel method called two-dimensional cell separation, which combines density gradient centrifugation and serial dilution. Through this two-dimensional cell separation approach, the microbial diversity captured with a single growth medium was 3–4-fold higher at the species level in comparison to traditional serial dilution methods. In addition, droplet-based techniques have been used to assist microbial isolation. For example, Afriza et al. developed a new anaerobic single-cell dispersion device that disperses droplets containing single cells into plates instead of spreading the samples by hand (Figure 1). Compared to the classical agar plate streak method, this approach significantly improved the efficiency of microbial isolation and cultivation by reducing the experimental time from around 17 days to 5 days while providing equivalent microbial diversity and relative abundance. Through this approach, 82 human gut bacterial species were obtained, including the first cultured member of 11 novel genera and 10 novel species that were fully characterized taxonomically [73]. The above two innovations are optimized nontargeted strategies, which are still labor- and resource-intensive. Next, we summarize targeted microbial isolation approaches that may be more likely to improve the efficiency of capturing microorganisms of interest.

#### 3.3.1. Live-FISH

Fluorescent in situ hybridization (FISH) is a classical fluorescent labeling method that designs fluorescent oligonucleotide hybridization probes, complementary to short sequence elements within the 16S rRNA common to phylogenetically coherent assemblages of microorganisms, to hybridize with suspensions of cells [74]. However, it requires the chemical fixation of the cells, which leads to the death of the labeled cells. Therefore, the labeled cells cannot be cultured. To address its shortcomings, Batani et al. [75] developed an improved FISH method called Live-FISH by removing the chemical fixation part of the standard FISH procedure. Notably, they achieved the sorting and culturing of live bacteria by combining Live-FISH with fluorescence-activated cell sorting (FACS). Next-generation sequencing technologies can help design species-level specific probes, which means that the Live-FISH method has a greater potential for targeted sorting of microorganisms of interest. However, the percentage of live cells obtained using this method is relatively low, and the rate of live cells needs to be improved.

#### 3.3.2. Metabolite Labeling

Monosaccharides and amino acids modified with fluorescent dyes can be assimilated by bacteria and then transformed into cell walls through bacterial metabolic processes, thus enabling the fluorescent labeling of bacteria. This method has little effect on the physiological processes of bacterial growth, metabolism, and reproduction [76]. However, a single fluorescent marker may not be highly discriminatory. Accordingly, Xu et al. [77] reported a novel method to enhance the accuracy of distinguishing specific species. On the basis of the differences in microbial metabolic characteristics, they labeled bacteria by applying multiple metabolites and modifying those metabolites with different reporters. In addition, Wang et al. [78] developed an off–on near-infrared (NIR) fluorescent probe 7-amino-1,3-dichloro-9,9-dimethylacridin-2(9H)one (DDAN), which conjugates pyroglutamic acid (recognition group) and DDAN (fluorophore). Given that pyroglutamyl aminopeptidase I (PGP-1), a bacterial hydrolase, can hydrolyze the amide bond of the l-pyroglutamate (L-pGlu) residue at the amino terminus of proteins and peptides, the DDPA can be applied for the visual sensing of PGP-1. As a result, they identified and cultivated eight bacteria strains with active PGP-1 by using DDPA. Due to the commonalities between different microorganisms, the specificity of this labeling method needs to be further enhanced.

#### 3.3.3. Antibody Labeling

Given that next-generation sequencing allows the identification of specific genes of microorganisms, antibody engineering may help to target the taxa of interest. Cross et al. [79] reported a novel microbial targeted sorting method called reverse genomics, which has successfully isolated three novel species from the human oral cavity. This method consists of the four steps. Firstly, sequencing is performed to predict the genes encoding epitopes of specific microorganisms. Next, the corresponding antigenic polypeptides are synthesized according to the genes. These antigenic peptides are then used to immunize animals to produce antibodies. Lastly, the antibody is coupled with fluorescein. These fluorescently labeled antibodies can then be used to label targeted microorganisms for subsequent sorting by FACS. The success of this approach relies on the accurate prediction of specific epitope genes, the labeling rate of fluorescent antibodies, and the robustness of the microbial culture. Although the reverse genomic is just used for oral microbial isolation, it offers a promising methodological reference for obtaining more microorganisms of interest from the human intestine.

Another antibody-based method is immunomagnetic separation. When superparamagnetic microbeads coated with antibodies are mixed with cells containing target substances, they can form a magnetic complex. Then, the complex can be manipulated by magnetic fields for sorting [80]. Although magnetic-assisted sorting is very widely used in the field of human cells, there are few cases where this method has been applied to sorting bacteria from the human gut. Notably, a recent study provided a method that can be a potential reference. Marcos et al. [81] prepared specific polyclonal antibodies against the surface of *Faecalibacterium prausnitzii* M21 and captured *Faecalibacterium prausnitzii* from the human intestinal microbiota by combining immunomagnetic separation with flow cytometry. However, it is important to note that the residue of immunomagnetic beads on the bacterial surface may lead to bacterial adhesion and, thus, affect the accuracy of flow sorting.

#### 3.3.4. Raman-Activated Microbial Cell Sorting (RACS)

Raman-activated microbial cell sorting (RACS) is another innovative approach for microbial isolation, which is based on isotope labeling [82]. Generally, stable isotope probes (SIPs) containing ^15^N, ^13^C, and ^2^H are designed to replace the original isotopes (^14^N, ^12^C, and ^1^H) in the DNA, RNA, or protein of the cell, thus labeling them. One of the effective stable isotope tools is D_2_O, which can enable the Raman detection of substrates shifted from the C–H band at 2000–2300 cm^−1^. The metabolic information of the microorganisms can be used to design a medium containing the specific metabolic substrate and D_2_O, which makes metabolic-activated bacteria generate a C–D band in the single-cell Raman spectrum. Then, these bacteria with characteristic peaks can be sorted by microfluidic devices and optical tweezers [82]. 

Several studies have shown the capacity of RACS for targeted sorting. Wang et al. [83] applied Raman microscopy and a deuterium isotope to sort out metabolic activated antibiotic-resistant bacteria (MA-ARB) in the intestinal microbiota of healthy adults. Lee et al. [84] developed a Raman-based high-precision (98.3% ± 1.7%) automated sorting platform that avoids the bias associated with manual sorting. Xu et al. [85] reported an optical tweezer-assisted single-cell sorting system that can accurately sort cells with diameters ranging from 1 to 40 µm, extending the applicability of RACS.

#### 3.3.5. Automated Microbiome Imaging and Isolation (CAMII)

In addition to labeling methods, morphology data of colonies can also assist in targeted microorganism sorting. As the colony morphology of certain microorganisms has distinctive characteristics, picking these colonies allows for obtaining their pure cultures. However, as the number of colonies is usually too large, this method may place a heavy burden on the experimenter. Moreover, the difference in the morphology of colonies may be tiny, which may also lead to bias when picking the target colonies. Interestingly, artificial intelligence techniques and automated equipment have led to a dramatic improvement in the method of targeted selection based on colony morphology. For example, Huang et al. [23] recently developed a machine learning approach called Automated Microbiome Imaging and Isolation (CAMII), which can discern nuanced features in multidimensional imaging and biological data (Figure 4B). On the basis of the obtained genomic and morphological information of the colonies, a machine-learning model can be trained to pick the specific genera. The researchers integrated an imaging system, an automated colony-picking system, a sequencing system, and a data analysis system in an anaerobic chamber. First, this device collects morphological data from colonies on plates, including density, size, convexity circularity, color, and inertia. Then, the morphological data are analyzed and the desired colonies are selected, which are then transferred to plates by an automated collection system to build a biobank. Next, the cultures in the biobank are sequenced and analyzed to obtain the corresponding genomic information. Lastly, morphological data and genomic information are combined for further in-depth screening. The CAMII system has over 20-fold higher isolation throughput than manual colony isolation by a person. This high-throughput automatic approach significantly reduces the labor burden of morphology-based isolation approaches, as well as provide a novel alternative to maximize the diversity of microorganisms isolated. 

#### 3.3.6. Gene-Targeted Microfluidic Isolation

Some microfluidic tools including iChip [86] and SlipChip [87,88] have been successfully applied to improve bacterial isolation. It is worth noting that Ma et al. conducted a genetically targeted microfluidics-based isolation and obtained a gut bacterium listed in the human microbiome project’s most wanted taxa [89] (Figure 4C). Firstly, researchers performed metagenomics sequencing of the samples to identify the target genes and designed PCR primers whilst they loaded bacterial suspension onto a microfluidic device to enable stochastic confinement of single cells. After several days of incubation, single colonies formed in each compartment of the chip. Then, the Slipchip was split, and the colonies in each compartment were randomly divided into two halves. On one half of the chip, target colonies were identified using PCR. Then, the target colony on the other half of the chip was retrieved for a scale-up culture. Lastly, the pure culture was sequenced to verify the isolation of the target microorganism. They slickly used the SlipChip to split the individual dispersed colonies into two halves for cultivation and detection, providing a valuable methodological reference for performing gene-targeted isolation of microorganisms. 

### 3.4. Cultivation

#### 3.4.1. Optimization of Medium

One important reason why microorganisms cannot be isolated and cultured is that the designed medium may not meet their metabolic requirements, such as a lack of nutrients or growth factors and inappropriate concentrations of certain components, oxygen, toxins, etc. [15]. Some researchers try to choose the most suitable medium for microbial culture by testing a large number of different conditions. Although this approach is time- and resource-consuming, many media suitable for microbial cultivation have eventually been developed. Notably, after cumulatively testing over 300 culture conditions, Diakite et al. [21] conducted a study to condense the culture conditions to 16 optimal types, which were sufficient to capture 98% of the total number of species previously isolated. However, blindly testing different media may not be enough to obtain certain hard-to-culture microorganisms.

Furthermore, metagenomics data can help researchers to obtain metabolic information about microorganisms including substrate utilization, oxygen demand, and antibiotic resistance [39]. For example, Rettedal et al. [90] determined the phylogenetic distribution of 16 antibiotic tolerance phenotypes in the human gut microbiota. With the guidance of high-throughput sequencing, the medium was modified. Then, they isolated and cultured two novel species with designed antibiotic combinations. Lugli et al. [91] used metagenomics data to predict four possible specific carbon sources for a certain gut microorganism, including arabinogalactan, pullulan, starch, and xylan. Next, they designed the medium using these sugars as the sole carbon source, and finally isolated and cultured two new strains of *Bifidobacterium*. Additionally, Yang et al. [92] applied metagenomics, pangenomics, and enzymology to identify the dominant glycoside hydrolase (GH) family of *Bifidobacterium*. As a result, they found that the application of cottonseed sugar, anhydrous D-alginate, and D (+)-cellobiose as the main carbon sources was beneficial to the growth of *Bifidobacterium*. 

#### 3.4.2. Coculture System

Some species that depend on other microorganisms for their survival can be cultured in coculture systems [93]. Accordingly, Tanaka and Traore et al. [94] designed a coculture device with a 0.2 μm membrane filter at the bottom, which was flanked by a soft agar medium with simple nutrient composition. Bacteria are inoculated in two layers of soft agar separated by a membrane filter to simulate interbacterial communication in vitro. This coculture device has successfully helped to isolate hitherto uncultured bacteria such as *Phascolarctobacterium* sp. BL377. Ge et al. [95] designed a coculture device called the nanoporous microscale microbial incubator (NMMI) to compartmentalize microorganisms in a large number of nanopores while ensuring that metabolites can circulate through the space. Thus, microorganisms requiring symbiotic factors can survive while being isolated. 

#### 3.4.3. Single-Cell Cultivation

Microfluidic devices have also been applied to improve microbial cultivation. For instance, Watterson et al. [96] used a droplet-based high-throughput culture method to screen antibiotic-resistant microorganisms in stool samples. Compared to conventional plates, this method showed a higher taxonomic richness and a larger representation of rare taxa. The technical key is the dispersion of bacteria in millions of picoliter droplets, which allows those slow-growing organisms to form colonies independently. Dennis et al. [97] conducted a similar study that used porous aluminum oxide chips containing mixed antibiotics for high-throughput culturing to screen the bacteria with antibiotic resistance. In addition, Ma et al. developed a chip wash device to monitor bacterial growth under different cultivation conditions [89]. This device allows for 3200 cultures at a time, with a Slipchip to confine single cells in each compartment. After the colonies are formed, the chip is slid again so that the microdroplets are reunited with the channel and washed into the collector with buffer. Then, the DNA in the collector is extracted for genetic analysis such as sequencing and PCR. They also tested the functionality of the chip wash method using a two-species model community to validate microbial growth on the chip. Lastly, they found a more than 1000-fold increment in DNA per strain in Slipchip compared to the DNA of the non-growth control, suggesting that this chip-washing method may be an efficient way to detect slow-growing bacteria.

## 4. Discussion

As microorganisms obtained from the human intestine have a wide range of applications in scientific research and industry, culturomics is continuously developing. However, to bring more human intestinal microorganisms into cultivation and improve efficiency, there are still many challenges. 

Due to the significant differences in microbial composition between fecal and intestinal, researchers need to choose an appropriate collection approach to access those mucus-associated microorganisms, which may be missed in fecal samples. For example, samples collected by multiple techniques may represent more diversity of the human gut microbes. Another alternative is to collaborate with clinicians, who may own more sample resources. The intelligent capsule seems to be a promising tool to collect samples mostly representing the originality of the human intestine. Despite its advantages, including precision and non-invasion over the ‘fecal way’ and endoscopy-assisted approaches, before intelligent capsules are widely used, it is necessary to reduce the cost of intelligent capsules and optimize their susceptible contamination. In addition to the collection methods, preservation methods are also worth noting, because they may influence the microbial originality in samples. Especially when investigators want to isolate some hard-to-culture microbes, they should preserve the raw samples carefully to minimize bias, which may occur in both cultivation and sequencing. 

Both culture-dependent and culture-independent approaches have improved the understanding of microorganisms and their relationship with the environment. High-throughput sequencing platforms, in particular, have rapidly enhanced our ability to understand the diversity of species in microbiomes. Metagenomics data have guided microbial isolation and cultivation such as medium and probe design [98,99,100,101]. Despite the vast database built, whether data ultimately help microbial cultivation still depends on the precise interpretation of the data. For unknown gut microorganisms, current bioinformatics tools may provide limited references. Assembly and phasing in heterogeneous environmental samples are also challenging. Therefore, in addition to overcoming the inherent shortcomings of culture-independent approaches, as a complementary approach to culture-independent approaches, culturomics is of note. Culturomics allows for the conservation of pure cultures that are necessary to directly assess the metabolic and physiological functions of microorganisms in culture-dependent studies. The current culture-dependent approaches are still being optimized, as reviewed herein, and the efficiency of isolation and culture of gut microorganisms may be enhanced in the future by merging both culture-dependent and culture-independent approaches.

Given that classical culturomics is labor- and resource-intensive, intelligent artificial techniques and robots may help address the shortcomings. A typical case is the so-called CAMII system based on automatic devices and machine learning, a high-throughput microbial culturomics approach. Although this method provides a paradigm for AI-assisted culturomics, its ability to tap into microbial resources remains to be further tested.

As a labeling method, in fluorescence-activated cell sorting (FACS), microorganisms are fluorescently labeled and sorted by flow cytometry. It is necessary to evaluate cellular autofluorescence interference, toxicity, and cell viability of the fluorescent dye, specificity of the fluorescent label, and prominence of the label, all of which may lead to failure of targeting. Therefore, to improve the efficiency of targeted sorting strategies, researchers need to ensure that labeling is specific and significant to avoid interference with microorganisms as much as possible. 

Additionally, it is worth trying to modify some emerging techniques to adapt to the specific requirements of gut microbial culture, such as an integration of sorting equipment with an anaerobic chamber. The methods that have been successfully applied in the field of nonintestinal microorganisms can also be tried in the exploitation of intestine microbial resources. The strategies and techniques discussed in this paper may provide some reference for further enriching the methodological library of gut microbial isolation and cultivation.

## 5. Conclusions

As branches of culturomics, nontargeted and targeted strategies have jointly helped to obtain hundreds of species of human gut microbes. Investigators have also optimized aspects of culturomics, namely, sample collection, sample processing, isolation, and cultivation. The optimizations have helped to achieve higher throughput, less labor burden, more precise manipulation, and richer diversity. However, the limitations of experimental devices and the lack of microbial information, such as the special physiological states (whether they are dormant or not), symbiotic and competitive information, and microbe–host interactions, may hinder the culture of parts of human intestinal microorganisms. Moreover, there are still many unknown constraints to be further explored and described. To break the bottleneck of current strategies for obtaining human intestinal microorganisms, researchers still need to creatively integrate multiple isolation and cultivation strategies with advanced techniques such as microfluidic and high-precision sorting, while maintaining an emphasis on in-depth information about the human gut microorganisms from both culture-dependent and culture-independent research.

## Figures and Tables

**Figure 1 microorganisms-11-01080-f001:**
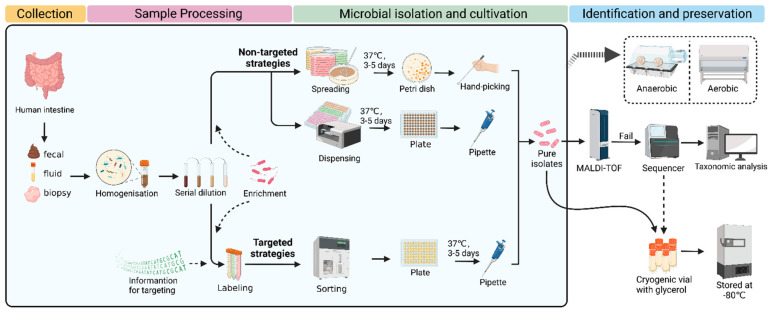
Schematic representation of nontargeted and targeted strategies for culturomics. The nontargeted strategies are usually to obtain as many species as possible in a sample, while the targeted strategies are more likely to sort certain microorganisms of interest. Created with BioRender.com.

**Figure 4 microorganisms-11-01080-f004:**
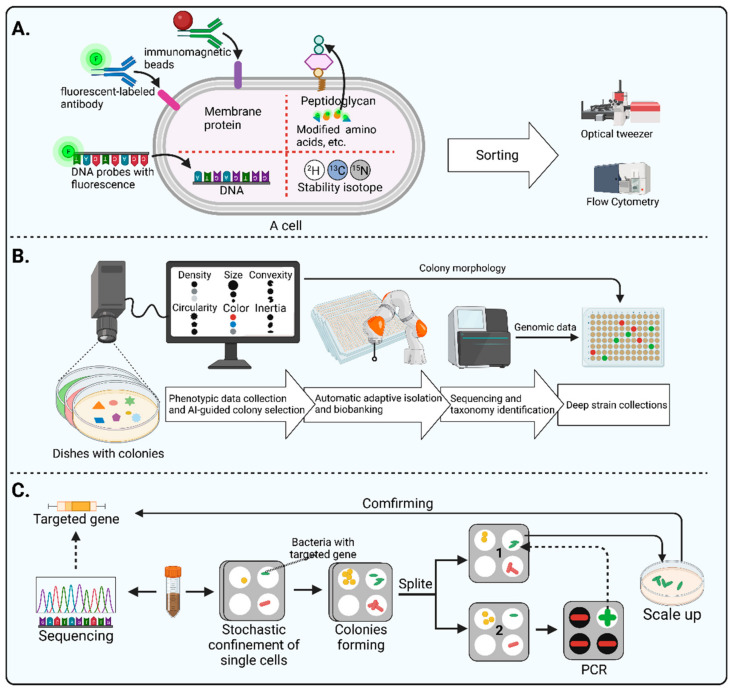
The summary of innovative approaches to isolate microorganisms. (**A**) Labeling-based method for cell sorting including Live-FISH, metabolites labeling, antibody engineering, and Raman-activated microbial cell sorting (RACS); (**B**) Automated Microbiome Imaging and Isolation (CAMII); (**C**) Gene-targeted microfluidic isolation. The 1 and 2 in the Figure 4C represent the top and bottom halves of the chip. Created with BioRender.com.

## Data Availability

No new data were created or analyzed in this study. Data sharing is not applicable to this article.

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
