# Peer review of "Isolation and Cultivation of Human Gut Microorganisms: A Review"

_microorganisms, 2023, doi:10.3390/microorganisms11041080_

Round 1

Reviewer 1 Report

The authors aimed to overview the strategies for obtaining human gut microbial resources, discussing the optimizations of parts of culturomics including sample collection, sample processing, microbial isolation and cultivation, pointing to provide a possible reference for developing better approaches to obtain microbial resources in the human gut.

The study covers some issues that have been overlooked in other similar topics. The structure of the manuscript appears adequate and well divided in the sections. Moreover, the study is easy to follow, but some issues should be improved. Some of the comments that would improve the overall quality of the study are:

I-) Authors must pay attention to the technical terms acronyms they used in the text.

II-) Conclusion Section: This paragraph required a general revision to eliminate redundant sentences and to add some "take-home message".

Minor editing of English language required.

Author Response

Response to Reviewer 1 Comments

Point 1: Authors must pay attention to the technical terms acronyms they used in the text.

Response 1: Thanks for your comment. We apologize for not being able to make every technical term acronyms clear. We have carefully checked technical terms acronyms used in the text and ensured that each key acronym was preceded by its corresponding full name at its first appearance.

Actions: As the reviewer suggested, We have added the full names of the following acronyms, including GI, NIR, DDAN, PGP-1, which are highlighted in yellow in the text.

  • gastrointestinal (GI)
  • near-infrared (NIR)
  • 7-amino-1,3-dichloro-9,9-dimethylacridin-2(9H)one (DDAN)
  • pyroglutamyl aminopeptidase I (PGP-1)

The above edits are in Line 127 (GI), Line 241 (NIR), Line 242 (DDAN), and Line 243 (PGP-1) of the revised manuscript, respectively.

Point 2: Conclusion Section: This paragraph required a general revision to eliminate redundant sentences and to add some "take-home message".

Response 2: Thanks for your suggestion. We have eliminated redundant sentences and streamlined statement in conclusion section to make this part clear and focused.

Actions: We reorganized the conclusion as you suggested to Lines 447-460 of the revised manuscript. The revised contents are as follows: “As the branches of culturomics, non-targeted, and targeted strategies have jointly helped to obtain hundreds of species of human gut microbes. Investigators have also optimized parts of culturomics, namely sample collection, sample processing, isolation, and cultivation. The optimizations have helped to achieve higher throughput, less la-bor burden, more precise manipulation, and richer diversity. However, the limits of experimental devices and lack of microbial information, such as the special physiological states (whether they are dormant or not), symbiotic and competitive information, and microbial–host interactions, may hinder the culture of parts of human intestinal microorganisms. Moreover, there are still many unknown constraints to be further explored and described. To break the bottleneck of current strategies for obtaining human intestinal microorganisms, researchers still need to creatively integrate multiple isolation and cultivation strategies and advanced techniques such as microfluidic and high-precision sorting, with an emphasis on in-depth information about the human gut microorganisms from both culture-dependent and culture-independent research.”

Reviewer 2 Report

It is an interesting review discussing different methods for isolation and cultivation the human gut microbiome. The authors provided nice figures summarizing different steps and approaches that used for this purposes. The review is well designed and easily flow.

One Minor point.

I would suggest the authors to include a comparison between culture based approaches and sequencing (molecular based) approaches  in human microbiome studies showing pos and neg in each one

Moderate language editing

Author Response

Response to Reviewer 2 Comments

Point 1: I would suggest the authors to include a comparison between culture based approaches and sequencing (molecular based) approaches in human microbiome studies showing pos and neg in each one.

Response 1: Thanks for your advice. As you mentioned, culture-based approaches and sequencing (molecular based) approaches are two fundamental and complementary strategies for human microbiome studies. Identifying their pos and neg will help to develop higher methods for gut microbiology research.

Actions: We have included the comparison of these two approaches in disscussion section as you suggested and mainly disscussed their features in microbial isolation and cultivation to Lines 411-426 of the revised manuscript, which has been highlighted in blue in the text. The revised contents are as follows: “Both culture-dependent and culture-independent approaches have improved the understanding of microorganisms and their relationship with the environment. The high-throughput sequencing platforms, in particular, have rapidly enhanced our ability to understand the diversity of species in microbiomes. Metagenomics data have guided microbial isolation and cultivation such as medium and probe designing [98–101]. Despite the vast database built, whether data ultimately help microbial cultiva-tion still depends on the precise interpretation of the data. For unknown gut microorganisms, current bioinformatics tools may provide limited references. Assembly and phasing in heterogeneous environmental samples are also challenging. Therefore, ex-cept for overcoming the inherent shortcomings of culture-independent approaches, as a complementary approach to culture-independent approaches, culturomics is of note. Culturomics allows for the conservation of pure cultures that are necessary to directly assess the metabolic and physiological functions of microorganisms in culture-dependent studies. The current culture-dependent approaches are still being optimized as this paper reviewed, and the efficiency of isolation and culture of gut microorganisms may be enhanced in the future by merging both culture-dependent and culture-independent approaches.”